# Influence of Equipment Operation Parameters on the Characteristics of a Track Produced with Construction 3D Printing

**Mikhail Elistratkin [1], Nataliya Alfimova [2,*], Daniil Podgornyi [1], Andrey Olisov [2], Vladimir Promakhov [2] and Natalia Kozhukhova [3]**

[1] Building Materials Science, Products and Structures Department, Belgorod State Technological University Named after V.G. Shukhov, 46 Kostyukova Str., 308012 Belgorod, Russia; mr.elistratkin@yandex.ru (M.E.); dan_podgor@mail.ru (D.P.)

[2] Research and Education Center «Additive Technologies», National Research Tomsk State University, 36 Lenin Ave., 634050 Tomsk, Russia; a.v.olisov@mail.tsu.ru (A.O.); vvpromakhov@mail.ru (V.P.)

[3] Department of Roads and Railways, Belgorod State Technological University Named after V.G. Shukhov, 46 Kostyukova Str., 308012 Belgorod, Russia; kozhuhovanata@yandex.ru

* Correspondence: alfimovan@mail.ru; Tel.: +7-9202002160

**Abstract:** Additive technologies are widely used in various industries. However, nowadays, the large-scale implementation of these technologies in the construction industry is difficult, due to a lot of open practical and scientific questions in terms of both building mixtures and 3D printing equipment. When performing studies focused on the development of cost-effective mixtures based on readily available raw materials for building extrusion 3D printing, it was found that the final result was determined by the rheology of the building mixture, the speed of the screw, and other factors. The article studied the combined effect on the extrusion of the building mixture and the parameters of the printed track of such factors as the thickness of the layer, the linear printhead traversed velocity of the forming device, and the speed of rotation of the screw. We aimed to establish relationships between the above factors, providing an increase in the stability of the printing process and the quality of the resulting structure. To carry out the research, an experimental program and original methods were developed, involving printing in different regimes using a laboratory construction 3D printer. Based on the regression analysis of the data obtained, it was found that the process of 3D printing by extrusion methods cannot be described by a linear function. It was found that a change in the linear speed of the nozzle movement can increase the yield of the mixture, and also lead to track stretching and the degradation of some parameters. The boundary value, in this case, is the layer thickness of 0.77–0.8 of the nozzle width. The response of the system to changes in the linear printhead traversed velocity and the frequency of rotation of the screw occurs in different ways. A change in the linear printhead traversed velocity at the optimal height of the layer has a slight effect on its width. Reducing the speed of rotation of the screw leads to a decrease in the overall dynamics of the mixture flow and an increase in its viscosity due to its thixotropic nature. When the previous speed of rotation of the mixture is restored, the dynamics of the flow are restored with a noticeable delay. In general, this is recommended to ensure the highest dynamics of the printing process. For the laboratory construction 3D printer and the building mixture used in the article, the regime with the following parameters was recommended: a linear printhead traversed velocity of 900 mm/min; an extruder frequency of 25 rpm; and a relative layer thickness of 0.8 (of the nozzle width). This regime provides the optimal ratio of performance/quality and the stability of track parameters.

**Keywords:** 3D printing; operation regime of construction 3D printer; parameters of the printing track; optimal thickness of the printed layer; control of construction 3D printing process

## 1. Introduction

Additive technologies, due to their numerous advantages [1,2], have found active practical application in various industries. The effectiveness of their application in such industries as medicine [3,4], electronics [5,6], aircraft [7,8], automobile [9,10], mechanical engineering [11,12], and instrumentation [13] has been proven many times. It is not surprising that the possibility of using additive technologies in the construction industry is of great interest to researchers and manufacturers [14–16]. Despite the current experience of the large-scale practical implementation of 3D printing [17], there are still many unresolved practical and scientific problems regarding building mixtures, as well as equipment for 3D printing [18].

An analysis of the current situation in the construction industry indicates that the technology of building construction using 3D printing methods has been formed as a result of the integration of the following three basic parameters:

- Constructive–structural, which determines the spatial scheme of printed elements and structures, represented as a digital model;
- Robotic, which includes control systems, movement, and the positioning of the molding device;
- Formular–technological, which includes the compositions of building mixtures, equipment, operations, and regimes for their preparation, molding, and providing conditions to achieve design properties.

From this point of view, construction 3D printing is an excellent example of trans-disciplinarity [19–21], the essence of which is as follows: within each individual subject area (building mixtures, equipment, building technologies), it is impossible to obtain a comprehensively effective technical solution. Therefore, the close interaction of all these areas with each other will achieve a synergistic effect as a result.

Previous studies [22–24] devoted to the development of cost-efficient building mixtures based on readily available raw materials for construction 3D printing using the extrusion method (i.e., contour printing) have found that, as a result of printing, the actual output of the mixture and the shape of the resulting tracks is determined not only by the rheology of the mixture and the speed of the screw, but also by other factors, such as layer thickness, linear printhead traversed velocity of the forming device, vibration, etc., which are interrelated. Therefore, changing them during the printing process greatly complicates the setup of a computer print management system, making it much more complicated in comparison to traditional fused deposition modeling (FDM) and its analogues. In this regard, of great practical interest is the study of the influence of the relationship between different factors on the 3D printing process and the result of extruding mixtures.

Many researchers have contributed to the study of this issue. Article [25] is devoted to a comprehensive analysis of construction 3D printing, and the great importance of the influence of printing parameters on the finished structure was identified. It has been found that, with an increase in nozzle diameter, it is possible to reduce the flowability of the mixture. An increase in printing speed can be achieved by increasing the flow of the mixture and the linear speed of the nozzle, and increasing the mobility of the mixture improves extrudability but reduces the maximum possible number of successive layers.

The authors of the study [26] analyzed the influence of the linear printhead traversed velocity of a rectangular printing nozzle with dimensions of 30 × 15 mm and the speed of extrusion of the mixture on the surface area of the resulting structure. Excessive extrusion speed (51.3 mL/s) and low linear printhead traversed velocity (60–80 mm/s)

created excess flow, resulting in material being pinched and bulging laterally. On the other hand, at extremely low mixture flow rates (37.9 mL/s) and high linear printhead traversed velocity (140–200 mm/s) of the nozzle, significant breaks in the printed track were observed. As a result, it was found that material consumption has the greatest influence on the parameters of the printed structure. It was also noted that the flow rate and linear printhead traversed velocity of the extruder are independent of each other.

Xiao I. et al. [27] note the particular importance of selecting a mixture for construction 3D printing. However, printing regime is also an important factor, which should be selected individually for different types of structures. Parameters such as the pumping distance, the size of the printed structures, the total volume of the concrete mix per cycle, the preparation time for printing, the size of the printed track, its possible deformation, the length of one printed layer, and the time intervals between layers will differ for different 3D printing conditions, such as laboratory, plant-manufactured, and in situ.

In the study of Kruger J. et al. [28], it was noted that a significant influence (the coefficient of variation was up to 30%) on the rheological parameters of the mixture and, as a result, on the accuracy of the adopted analytical model, was exerted by the frequency of stirring the concrete mixture. Additionally, during the experiment, according to the theoretical model, the optimal printing speed (87 mm/s) and layer thickness (8 mm) were determined with the following experimental parameters: the layer width was 30 mm, nozzle diameter was 25 mm, and layer height variation was 8–15 mm. This indicates the absence of a direct correlation between the parameters obtained with the rheometer and the printability of the extruder. This is an urgent problem and requires further research.

Xiao, J. and Zou, S., et al. [29] analyzed the parameters that affect the quality of the printed track, the most important of which were identified as the following: material consumption in the nozzle (Q), layer width (w), layer height (h), and linear printhead traversed velocity of the nozzle (v). Theoretically, the formula describing the dependence of these parameters is as follows:

$$Q = w \cdot v \cdot h$$

For any of the given values with an acceptable accuracy, this allows the determination of the remaining values. Thus, having a stable flow rate of the mixture, a constant nozzle diameter and a certain printing speed, it is possible to choose the optimal layer height, since at an excessively low layer height significant spread occurs with substandard deformations or ruptures. However, with excessive layer height, the stability of the structure is significantly reduced. Thus, there is an optimal value for the ratio of the height of the layer to its width.

The work of Liu, Z. [30] is devoted to studying the influence of printing parameters, such as nozzle speed $\zeta$, corner radius R, and nozzle aspect ratio $\varphi$. A three-dimensional numerical model was developed by solving the Navier–Stokes equations. The Bingham model was used to characterize the properties of materials during 3D printing. The results obtained indicated a non-regular mass distribution of the printed track on the turning sections. At the same time, a low influence of rheological properties on this phenomenon was observed. A mathematical model was obtained for the uniformity of the mixture mass distribution in the track with the following equation:

$$\Phi = 1.23 - 0.045R + 0.034\varphi - 0.024\zeta + 8.2 \times 10^{-3}R \cdot \varphi$$

The mass distribution coefficient is mainly influenced by the printing parameters, and not by the properties of the mixture. To provide more homogeneous mass distribution at the corners, it is recommended to use printing regimes at higher linear speeds while increasing the corner radius and nozzle sides ratio. Thus, based on the presented data in [30], it can be seen that the issue of the stability of the extrusion process, and the search for effective tools for managing and optimizing print parameters, is currently still far from a final solution. There are no universal mathematical models that are suitable for practical use and adequately describe the process of forming the parameters of a printed

track. A lot of researchers believe that the printing parameters have a decisive influence on the performance of the resulting tracks. At the same time, other researchers consider the rheology of mixtures to be an equivalent or basic factor. In many works, it is possible to simplify the system under study by using a number of parameters as a constant and studying the remaining ones. This approach is justified in laboratory conditions, but is unacceptable in the practical implementation of the technology.

In this regard, the purpose of this article was to study the combined effect of the layer thickness, the linear printhead traversed velocity of the forming device, and the speed of rotation of the screw on the extrusion of the mixture and the parameters of the printed track. This will allow the development of recommendations for choosing regimes that provide an increase in the stability of the process and the quality of the resulting structure.

## 2. Materials and Methods

### 2.1. Materials

To prepare the mixtures, the following raw materials were used: Portland cement CEM II 42.5 S; dry purified two-fraction quartz sand with a grain size of 0.315–1.25 mm and with fineness modulus of 1.4; polycarboxylate superplasticizer PC type S; air-entraining agent Poliplast Aero 815; tap water.

The process of preparing the mixture consisted of mixing 1 part of Portland cement with 4 parts of sand (by wt.). Then, 90% mixing water with a plasticizer was introduced into the mixture; the water–cement ratio (W/C) was 0.74 (in commercial concentration). After curing for 5–10 min, the remaining water and an air-entraining additive-plasticizer were added into the mixture, and intensive mixing was carried out. The resulting concrete at the age of 28 days with an average density of 1850–1900 kg/m$^3$ provided a compressive strength of 12–15 MPa.

### 2.2. Experimental Setup and Equipment

Experimental studies were carried out using a laboratory 3D printer (Figure 1).

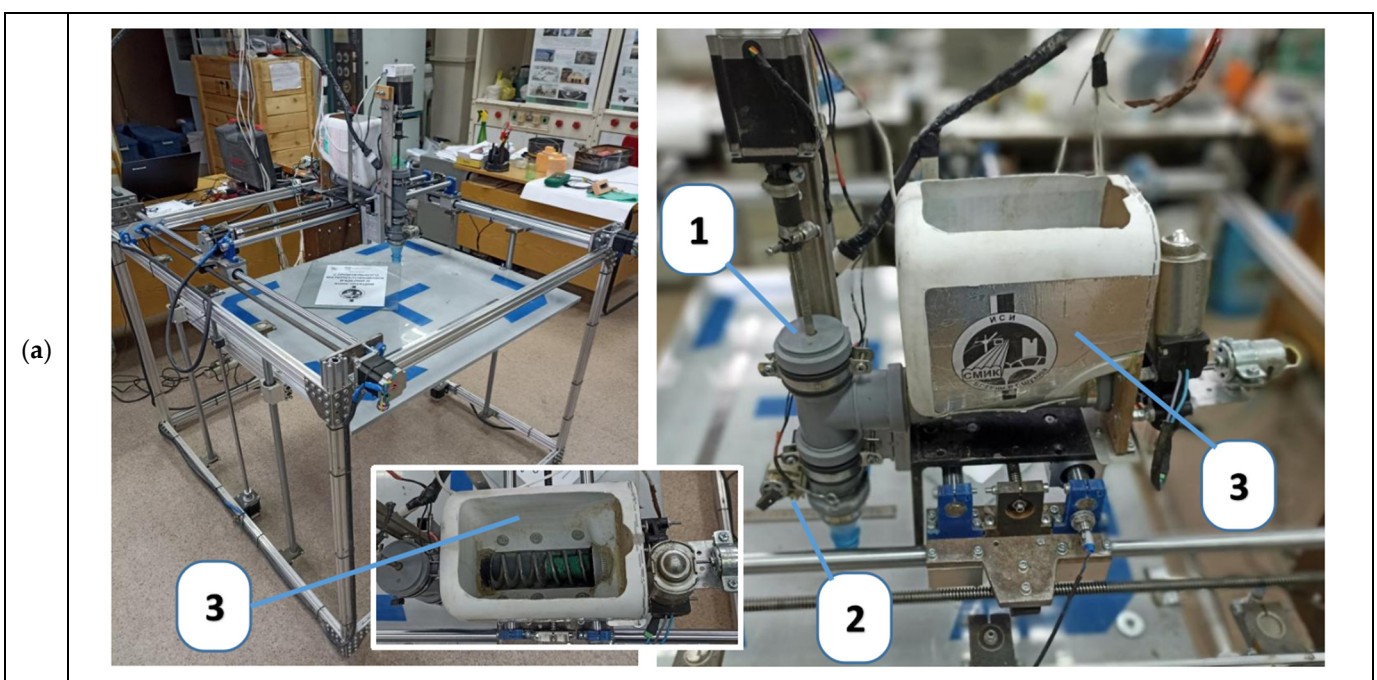

(a)

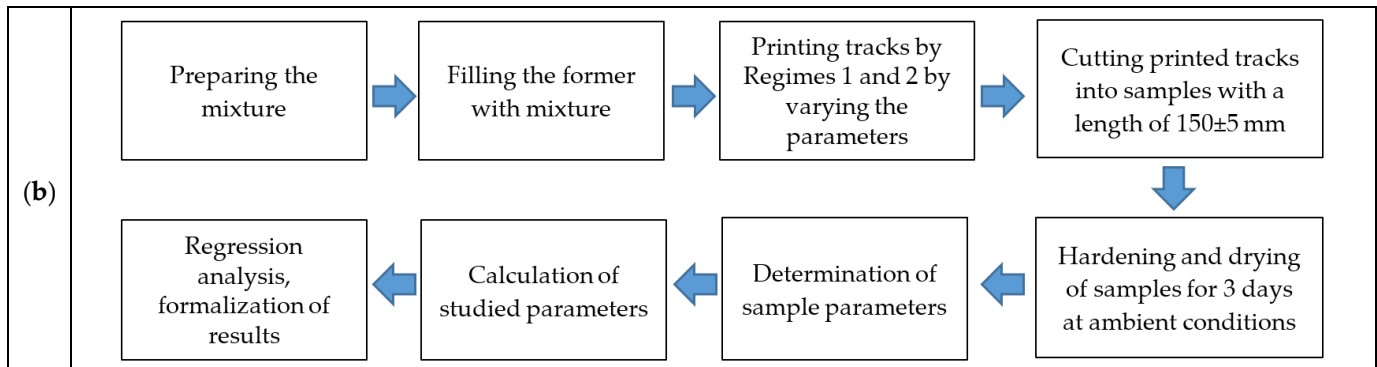

**Figure 1.** Laboratory 3D printer, designed in Belgorod State Technological University. V.G. Shukhov (Belgorod, Russia): 1—vertical screw blower; 2—hinged manually regulable vibrator; 3—supply bin-feeder: (**a**) general view of the 3D printer; (**b**) flow chart of experimental method.

The configuration of the forming device was upgraded to ensure the stable supply of mixture with different viscosity regardless of the mixture amount in the bin. The modernization consisted of the following procedure: the mixture supply path was divided into two modules. The first module was a vertical screw blower (1) controlled by the printer controller with a mounted manually regulable vibrator (2) on the nozzle. The second one was a supply bin-feeder (3) with a volume of 5 dm³, which pumped the mixture into the extruder due to a horizontal vibro-screw feed, with manual analog regulation of the regime. During the preliminary experiments, the feeder operation regime was selected and was kept constant during the research. The vibrator fixed on the nozzle was not used during the experiment. Its presence in the design of the forming device was necessary for the implementation of service regimes (filling and unloading the system), the stabilization of the 3D printing process when resuming after breaks lasting more than 2–3 min, and for working with low-plastic and highly thixotropic mixtures, as well.

The working area of the 3D printer plate was 75 × 75 cm; thus, it was able to print objects up to 80 cm in height. The maximum linear speed of the forming device along the X and Y axes was 1000 mm/min (limited by software), and extruder screw rotation speed was up to 60 rpm.

*2.3. Methods*

To study the influence of such factors as extrusion speed ($X2$) and linear printhead traversed velocity ($X3$) on the 3D printing process, a G-code generator was developed in the form of a spreadsheet. This was highly convenient for the varying of different parameters. The test sample consisted of 3 series of linear segments with different layer heights ($X1$): 10, 15, and 20 mm, printed at a constant nozzle diameter of 20 mm. For each selected layer height, 2 parallel linear segments were printed, using two different regimes, as follows:

- Regime 1: constant speed of rotation of the screw, with the linear printhead traversed velocity of the forming device varying at three levels;
- Regime 2: constant linear printhead traversed velocity of the forming device with the rotation speed of the screw varying at three levels.

The list of combinations of the studied factors (variables) in natural and coded form is presented in Table 1.

**Table 1.** Combinations of the studied factors.

| No | Sample ID | Regime | Studied Factors (Variables) in Natural and Coded Form | | | Variation Levels of Studied Factors (Variables) | | | | | |
| --- | --- | --- | --- | --- | --- | --- | --- | --- | --- | --- | --- |
| | | | X1: Layer Height h, mm | X2: Extrusion Speed, rpm | X3: Linear Printhead Traversed Velocity, mm/min | X1 | X2 | X3 | $X1^2$ | $X2^2$ | $X3^2$ |
| 1 | 1-1 | | 15 | 30 | 1000 | 0 | 1 | 1 | 0 | 1 | 1 |
| 2 | 1-2 | 1 | 15 | 30 | 750 | 0 | 1 | 0 | 0 | 1 | 0 |
| 3 | 1-3 | | 15 | 30 | 500 | 0 | 1 | −1 | 0 | 1 | 1 |
| 4 | 2-1 | | 15 | 10 | 1000 | 0 | −1 | 1 | 0 | 1 | 1 |
| 5 | 2-2 | 2 | 15 | 20 | 1000 | 0 | 0 | 1 | 0 | 0 | 1 |
| 6 | 2-3 | | 15 | 30 | 1000 | 0 | 1 | 1 | 0 | 1 | 1 |
| 7 | 3-1 | | 10 | 30 | 1000 | −1 | 1 | 1 | 1 | 1 | 1 |
| 8 | 3-2 | 1 | 10 | 30 | 750 | −1 | 1 | 0 | 1 | 1 | 0 |
| 9 | 3-3 | | 10 | 30 | 500 | −1 | 1 | −1 | 1 | 1 | 1 |
| 10 | 4-1 | | 10 | 10 | 1000 | −1 | −1 | 1 | 1 | 1 | 1 |
| 11 | 4-2 | 2 | 10 | 20 | 1000 | −1 | 0 | 1 | 1 | 0 | 1 |
| 12 | 4-3 | | 10 | 30 | 1000 | −1 | 1 | 1 | 1 | 1 | 1 |
| 13 | 5-1 | | 20 | 30 | 1000 | 1 | 1 | 1 | 1 | 1 | 1 |
| 14 | 5-2 | 1 | 20 | 30 | 750 | 1 | 1 | 0 | 1 | 1 | 0 |
| 15 | 5-3 | | 20 | 30 | 500 | 1 | 1 | −1 | 1 | 1 | 1 |
| 16 | 6-1 | | 20 | 10 | 1000 | 1 | −1 | 1 | 1 | 1 | 1 |
| 17 | 6-2 | 2 | 20 | 20 | 1000 | 1 | 0 | 1 | 1 | 0 | 1 |
| 18 | 6-3 | | 20 | 30 | 1000 | 1 | 1 | 1 | 1 | 1 | 1 |

The factors we used largely covered the most effective printing regimes that can be applied with a laboratory 3D printer.

Within the framework of the experiment, such parameters of the experimental tracks as: *input*, *studied*, *technical* and *nominal numeric* were evaluated.

The *input* parameters were the following:

- Layer height (X1) is the height of the printed layer (cm), which ranged from 1 to 2 cm in the article. Increasing the layer height leads to an increase in the height of the final structure and reduces the number of horizontal seams that weaken the structure. However, when printing a structure with a varying cross-section, when the longitudinal axes of the layers are displaced relative to each other, and a large layer thickness can lead to a loss of stability of the freshly printed structure;
- Extrusion speed (X2) is the rotation speed of the extruder screw, which ranged from 10 to 30 rpm in the article. This parameter largely determines the amount of mixture output and the width of the printed track;
- Linear printhead traversed velocity (X3) is the speed at which the extruder nozzle moves during printing, which ranged from 500 to 1000 mm/min in the article. Increasing this parameter contributes to an increase in the overall speed of the structure formation, reduces the effect of printer structure vibrations on the straightness of the track.

The *studied* parameters were the following:

- Y1 is the maximum track width (MTW), in mm. It is determined by the top layer, during the printing process in which there is no influence of the glass substrate. This parameter is the most visually significant in assessing the stability of the printing process. At the same time, printing a track with a variable width can be used as a way to increase the aesthetic expressiveness of the design. There is no consensus on the best track width in relation to nozzle diameter. Within the framework of this article, the range of 25-40 mm was identified as optimal;
- Y2 is specific consumption of the "concrete mixture/layer thickness" ratio (SCCM/LT), in $g/cm^2$. It represents the ratio of the specific consumption of the mixture for printing a track related to the layer thickness. A specific case of construction 3D printing should correspond to its own range of optimal values of this property, which makes it extremely difficult to compare the values obtained by different researchers. In this case, the optimal variation range was 4–7 $g/cm^2$;
- Y3 is the section utilization factor (SUF). This is a dimensionless parameter. It is calculated as a "working section area of the layer/sectional area of the layer" ratio (WSAL/SAL) and characterizes the part of the mixture in the track involved in receiving and transmitting loads. Considering this parameter, it is recommended to focus on the maximum approximation of its values to 1. This corresponds to a strictly rectangular cross-sectional shape of the track and a smooth outer surface of the structures. In practice, the specified value when using a round nozzle is unattainable; therefore, within the framework of the study, coefficient values of 0.8, at least, were recognized as acceptable.

The *technical* parameters were as follows:

- Contact area width (CAW), in mm. This parameter is used to calculate other parameters;
- Specific consumption of the concrete mix (SCCM), in g/cm. This parameter is calculated as the ratio of the mass of the sample to the product of its length and the number of layers;
- Working section area of the layer (WSAL), in $cm^2$. This parameter is calculated as the product of the width of the contact zone of the layers and the height of the layer;
- Sectional area of the layer (SAL), in $cm^2$. The track cross-sectional area is approximately determined as the area of an ellipse with a small diameter equal to the layer height, and a large diameter equal to the maximum track width (Y1);

The *nominal numeric* parameters were as follows:

- Visual assessment of results (VAR). This is evaluated in points from 0 to 5 and characterizes the overall aesthetics of the sample. This assessment is not completely objective and is intended only to show the priorities and preferences of the authors of the article.

The test sample was printed on a glass substrate in two layers in the forward and reverse directions. A general view of the test sample and different stages of its manufacturing process is shown in Figure 2.

From each freshly printed track with a length of 20 cm, produced according to the corresponding regime, a section with a length of about 15 cm was cut. After curing for 3 days in air dry conditions, the length, weight, and width of the contact zone of the layers were measured for each sample. A visual evaluation of the resulting structure was also carried out. The experiment was independently repeated 3 times.

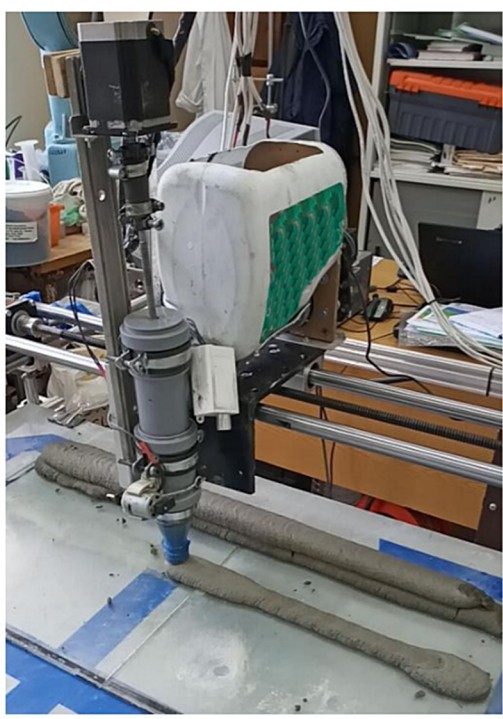

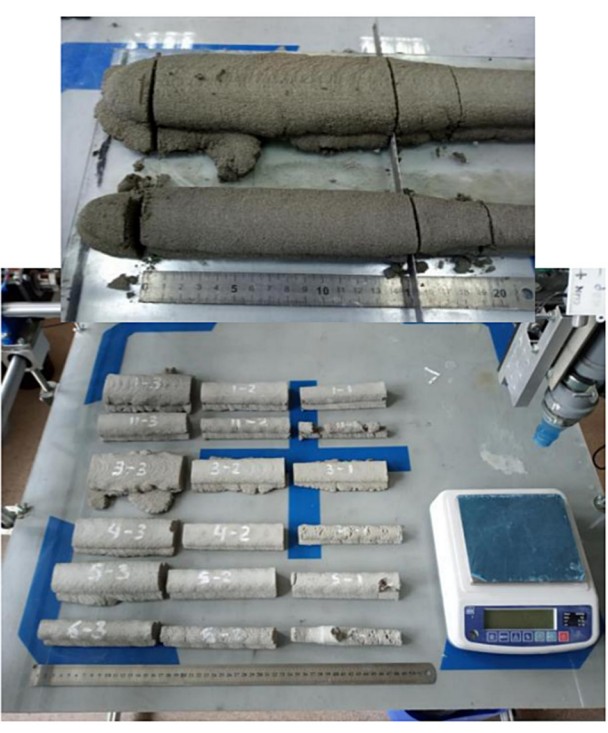

(**a**)                                                            (**b**)

**Figure 2.** Printing process and preparation of test samples: (**a**) track printing process; (**b**) cutting the track into samples with a length of ≈15 cm, and a general view of the samples prepared for the experiment.

## 3. Results

In accordance with the experimental design presented in Table 1, which indicates the studied options for the combinations of the studied printing parameters, such as linear printhead traversed velocity (X3), extrusion speed (X2), and layer height (X1), the basic output parameters (Y1–Y3) of the printed layer were obtained. The basic parameters of test samples are presented in Table 2.

**Table 2.** Basic parameters of test samples.

| No | Sample Code | Sample Weight, g | Sample Length, mm | Contact Area Width, (CAW), mm | Y1: Maximum Track Width, (MTW), mm | Specific Consumption of Concrete Mixture, (SCCM), g/cm | Y2: Specific Consumption of Concrete Mixture/Layer Thickness (SCCM/LT), g/cm² | Working Section Area of the Layer (WSAL), cm² | Sectional Area of the Layer (SAL), cm² | Y3: Section Utilization Factor, SUF | Visual Assessment of Results, VAR |
|---|---|---|---|---|---|---|---|---|---|---|---|
| 1 | 1-1 | 227.7 | 148 | 29 | 35 | 7.69 | 5.13 | 4.35 | 5.1 | 0.86 | 5 |
| 2 | 1-2 | 311.3 | 149 | 40 | 45 | 10.45 | 6.97 | 6 | 6.6 | 0.91 | 4 |
| 3 | 1-3 | 475.3 | 148 | 55 | 65 | 16.06 | 10.71 | 8.25 | 9.4 | 0.87 | 2 |
| 4 | 2-1 | 87.2 | 145 | 10 | 19 | 3.01 | 2.01 | 1.5 | 2.6 | 0.59 | 0 |
| 5 | 2-2 | 171.1 | 147 | 20 | 25 | 5.82 | 3.88 | 3 | 3.6 | 0.84 | 3 |
| 6 | 2-3 | 271.9 | 146 | 33 | 43 | 9.31 | 6.21 | 4.95 | 6.1 | 0.81 | 5 |
| 7 | 3-1 | 180.6 | 147 | 32 | 35 | 6.14 | 4.09 | 3.2 | 3.4 | 0.93 | 3 |
| 8 | 3-2 | 238.1 | 147 | 41 | 44 | 8.1 | 5.4 | 4.1 | 4.3 | 0.94 | 3 |
| 9 | 3-3 | 372.5 | 145 | 60 | 74 | 12.84 | 8.56 | 6 | 7.1 | 0.85 | 1 |
| 10 | 4-1 | 100.3 | 150 | 18 | 22 | 3.34 | 2.23 | 1.8 | 2.1 | 0.85 | 1 |
| 11 | 4-2 | 173.6 | 148 | 30 | 38 | 5.86 | 3.91 | 3 | 3.6 | 0.83 | 5 |
| 12 | 4-3 | 246.3 | 147 | 42 | 49 | 8.38 | 5.59 | 4.2 | 4.8 | 0.88 | 4 |
| 13 | 5-1 | 213.6 | 147 | 21 | 27 | 7.27 | 4.85 | 4.2 | 5.1 | 0.82 | 4 |
| 14 | 5-2 | 272.9 | 148 | 26 | 32 | 9.22 | 6.15 | 5.2 | 6.1 | 0.85 | 5 |
| 15 | 5-3 | 389.8 | 146 | 35 | 44 | 13.35 | 8.9 | 7 | 8.4 | 0.83 | 3 |
| 16 | 6-1 | 76.9 | 150 | 12 | 21 | 2.56 | 1.71 | 2.4 | 3.8 | 0.63 | 0 |
| 17 | 6-2 | 165 | 148 | 14 | 20 | 5.57 | 3.71 | 2.8 | 3.7 | 0.75 | 2 |
| 18 | 6-3 | 233.5 | 150 | 19 | 27 | 7.78 | 5.19 | 3.8 | 5.1 | 0.75 | 3 |

The resulting data set for the most significant characteristics of the test samples was subjected to regression analysis using the variables presented in Table 1. As a result, the following equations were obtained:

- for layer width:

$$Y1 = 33.7 - 7.6X1 + 7.7X2 - 12.5X3 - 2.6X1^2 + 0.7X2^2 + 8.2X3^2 \text{ (mm)} \quad (1)$$

with a standard deviation of 0.95;

- for the specific consumption of the mixture per layer related to its thickness:

$$Y2 = 5.36 + 0.06X1 + 1.60X2 - 2.11X3 - 0.79X1^2 - 0.25X2^2 + 1.11X3^2 \text{ [((g/cm)/cm)}] \quad (2)$$

with a standard deviation of 0.98;

- for the section utilization factor:

$$Y3 = 0.86 - 0.05X1 + 0.06X2 - 0.004X3 + 0.01X1^2 - 0.04X2^2 - 0.05X3^2 \quad (3)$$

with a standard deviation of 0.88.

For better visual perception, nomograms were plotted (Figures 3–5) on the basis of the obtained Equations (1)–(3). The length of all samples presented in Figures 3–5 is 150 ± 5 mm. Detailed characteristics of all the parameters are shown in Table 2.

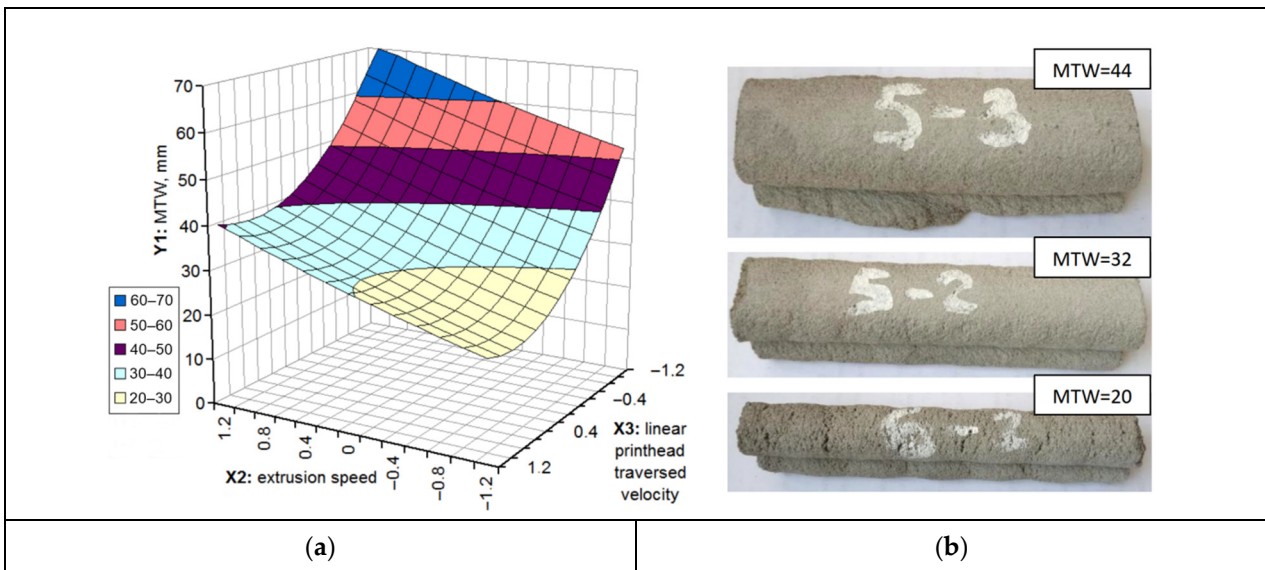

| (a) | (b) |

**Figure 3.** Effect of screw speed and linear printhead traversed velocity of the former on track width Y1 (when X1 = 0): (**a**) graphic representation of relationship between screw speed and linear printhead traversed velocity on track width; (**b**) visual assessment of MTW effect on the samples view. The length of the samples is 150 ± 5 mm.

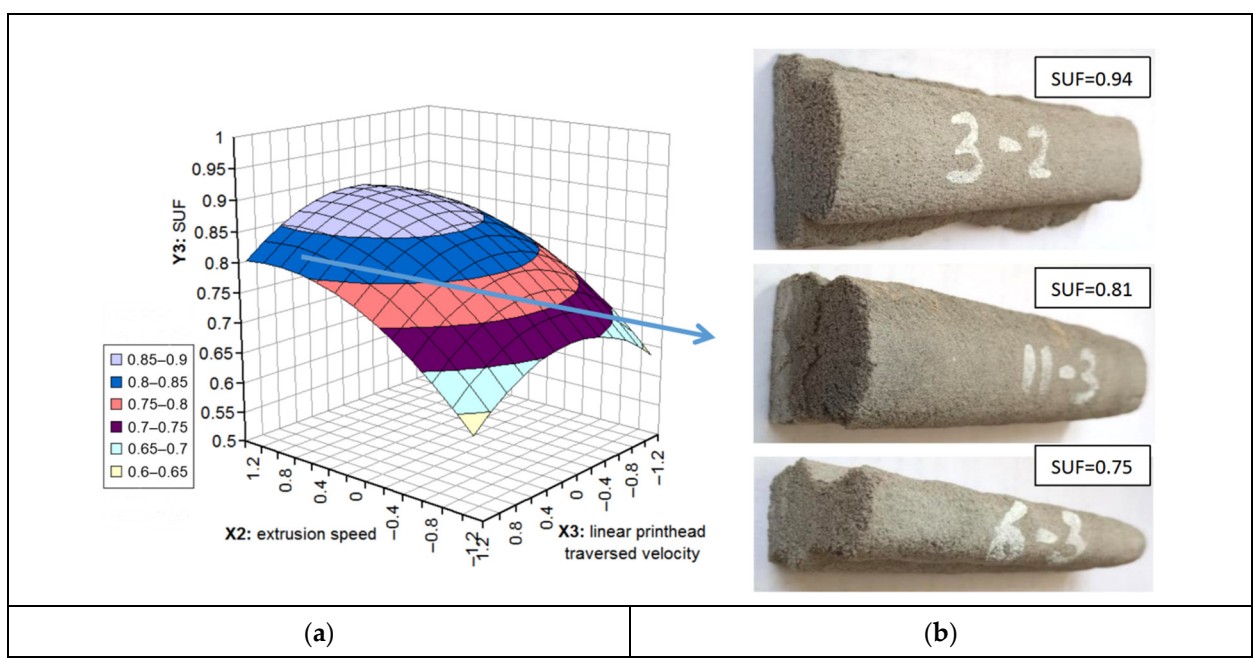

**Figure 4.** Effect of the screw speed and linear printhead traversed velocity of the former Y2 on the specific consumption of the mixture related to the layer thickness (at X1 = 0). (**a**) graphic representation of relationship between screw speed, linear printhead traversed velocity and specific consumption of the mixture; (**b**) visual assessment of SCCM/LT effect on the samples view. The length of the samples is 150 ± 5 mm.

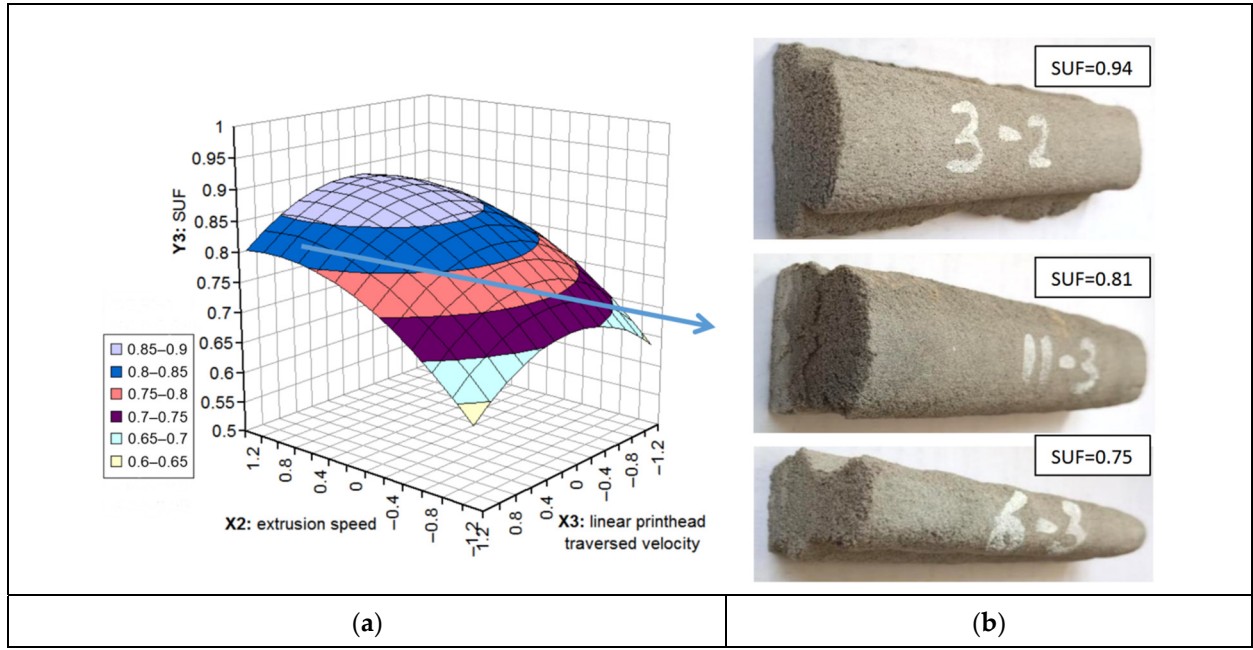

**Figure 5.** The effect of the screw speed and the linear printhead traversed velocity of the forming device on the utilization factor of the section Y3 (when X1 = 0): (**a**) graphic representation of relationship between screw speed, the linear printhead traversed velocity and utilization factor; (**b**) visual assessment of SUF effect on the samples view. The length of the samples is 150 ± 5 mm.

The nomograms shown in Figures 3–5 correspond to a certain layer height. Images for other values of the layer height are not shown here, but can be easily obtained based on Equations (1)–(3).

## 4. Discussion

An increase in the layer height (X1), as is commonly believed, should reduce the resistance to the mixture output, i.e., increase the value of the parameter Y1. This is due to the fact that the mixture from the nozzle does not exit into the free volume, but almost always rests against the substrate or the underlying layer. Due to the excess pressure caused by the extruder, the mixture undergoes horizontal deformations. As a result, the track width is larger than the nozzle width. This factor, when using simple systems with a non-rotating round nozzle, is positive, since it contributes to an increase in the width of the contact zone of the layers and increases the efficiency of the section.

The vertical pressure of the mixture leaving the nozzle, on the one hand, improves the connection of the layers and, on the other hand, creates an additional load on the base at the current printing point. This phenomenon also reduces the performance of the extruder, which can be used as a conventional pump whose performance decreases with increasing pressure. With a higher layer, the resistance to the lateral movement of the mixture and the output pressure from the nozzle decreases, which should lead to an increase in flow rate and all other components remaining equal.

However, according to the data shown in Figure 6A, plotted on the basis of the data of Table 2, in Regime 1 (the speed of the extruder is constant), the output of the mixture increased only until a relative thickness of 0.77–0.80 of the nozzle width was reached. The greater the increase in layer thickness, the lower the linear printhead traversed velocity of the nozzle, incurring decreases to the mass yield of the mixture. This phenomenon cannot be explained by reaching the maximum output of the extruder, since there would be a flattening of the curve on the graph (Figure 6).

A possible reason for the decrease in the yield of the mixture with an increase in the thickness of the layer above a certain threshold is the decrease in the efficiency of the screw extruder in the absence of the back pressure of the mixture at the output.

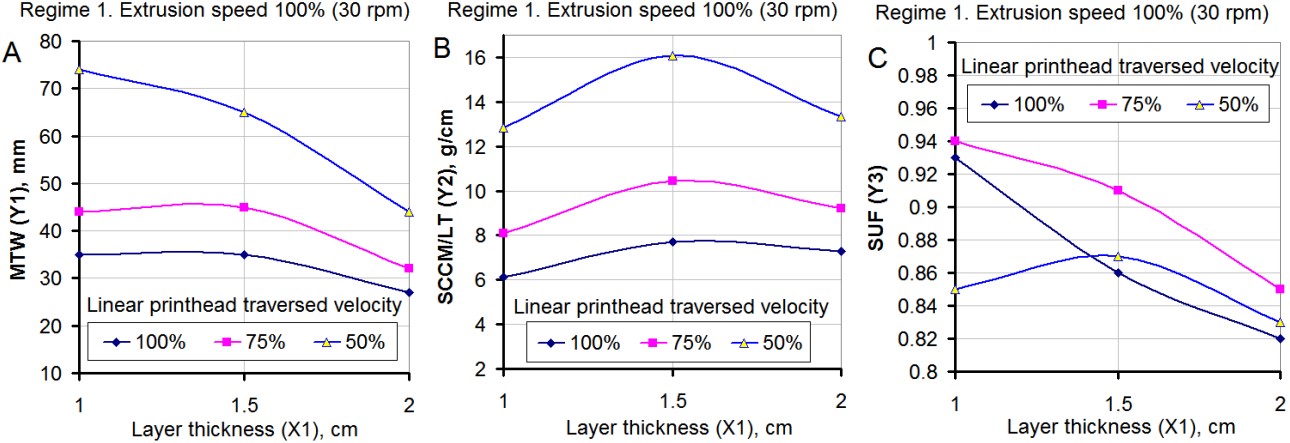

**Figure 6.** Influence of layer thickness on different print parameters when extrusion speed is constant (Regime 1: (**A**) maximum track width; (**B**) specific consumption of the "concrete mixture/layer thickness" ratio; (**C**) section utilization factor

Another parameter to note is that the yield of the mixture from the nozzle is provided not only by ejection by the screw, but also by the elongation of the mass having a high cohesion, due to the translational movement of the forming device.

Figure 6A shows that, for the linear printhead traversed velocity of 50 and 75% of the maximum value, the width of the printed track remains almost constant in the range of its relative height of 0.5–0.8 and then begins to decrease, which is caused by the narrowing of the track when it is stretched.

The section utilization coefficient also tended to decrease steadily with increasing layer thickness (Figure 6C). The highest values of this parameter were achieved at a linear printhead traversed velocity of the nozzle of 75% of the maximum. In this case, the

printed track had the smallest barrel shape, and the printed wall had the largest section capable of receiving loads.

The stated assumption well explains the progressive drop in the productivity of the extruder with an increase in the layer thickness and a decrease in the linear printhead traversed velocity. On the other hand, it allows substantiating the optimal relative layer thickness for the accepted printing method of 0.75–0.8 of the nozzle width. However, additional studies are required to confirm the validity of this statement when using mixtures with a different rheology or extrusion method.

Regime 2 (constant maximum linear printhead traversed velocity) is characterized by a reduced extruder productivity compared to a Regime 1, and, accordingly, lower overall dynamics of the movement of the mixture in the forming device. According to studies [31–33], this factor is also significant for the results of construction 3D printing. The difference in the yield of the mixture for samples 1-1, 2-3, 3-1, 4-3, 5-1 and 6-3, printed at the same settings but in different regimes, reached 30–35% by wt.

This can be explained by the fact that, in Regime 1, the intensity of the action of the screw on the thixotropic mixture was constant and maximum. In Regime 2, maximum screw speed and nozzle movement were accelerated step by step. A decrease in the viscosity of the mixture due to an increase in the intensity of the mechanical influences probably did not have time to occur during the printing of the test section. This led to an increase in the SCCM/LT parameter. This aspect is a serious complicating factor from the point of view of development a program for controlling the printing process. Thus, it is desirable to provide a temporary decrease in the output of the mixture (for example, when passing corners, pairing tracks, etc.) by increasing the linear printhead traversed velocity and not by breaking the screw.

The dependence of the mixture yield on the layer thickness for samples 2-2, 4-2, 6-2 (Figure 7B) was constant. This was probably due to the "pulling effect" of the mixture from the nozzle during its linear movement, as described earlier. This was accompanied by an almost linear change in the layer width and an increasing cross-section utilization coefficient (Figure 7C) in the range of the relative layer thickness of 0.5–0.7. At the same time, the absolute values of the layer width were in the optimal range of 25–40 mm, and the resulting tracks had the highest visual assessment of 4–5 points (Table 2).

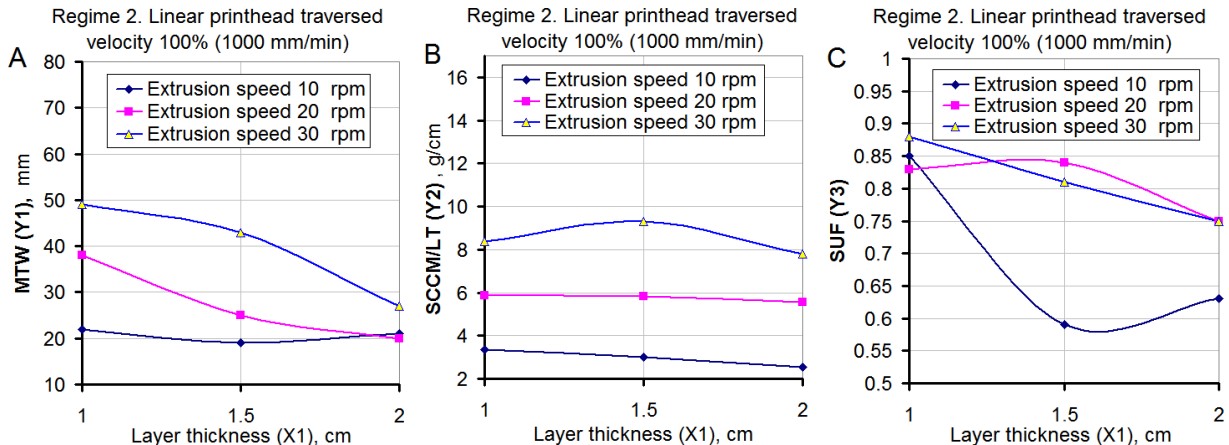

**Figure 7.** Influence of layer thickness on different print parameters when the linear printhead traversed velocity is constant (Regime 2): (**A**) maximum track width; (**B**) specific consumption of the "concrete mixture/layer thickness" ratio; (**C**) section utilization factor.

The regime with the lowest screw rotation speed (10 rpm) turned out to be clearly inappropriate for the high speed of the nozzle movement, which led to an unsatisfactory condition of the tracks, up to their breaks. It should be noted that, when a rupture occurs, the yield of the mixture drops sharply, despite the fact that a continuous track is formed with defects and a cross section smaller than the nozzle diameter prior to this. The re-

sumption of printing occurs only when the amount of mixture supplied from the nozzle is sufficient for it to engage with the bottom layer. This, once again, confirms the above assumption about the role of the "effect of pulling" the mixture from the nozzle during its movement.

To determine optimal printing regimes, criteria should be clarified. Most visually noticeable is the change in layer width (MTW). From a structural point of view, the value of the section utilization factor (SUF) is important. From the position of fabricability, the maximum printing speed is important.

Based on a combination of factors, the set of rational printing regimes for the optimal layer thickness is within the selected area (Figure 8). At the same time, to ensure maximum performance, it is recommended to select points with a higher linear printhead traversed velocity. For the passage of curved sections, it is advisable to slightly lower the linear printhead traversed velocity without changing the speed of the extruder. This will avoid pulling the mixture layer towards the center of the rounding. At the same time, the track width should not change significantly, since the decrease in the "pulling effect" will be compensated by the unchanged regime of the extruder.

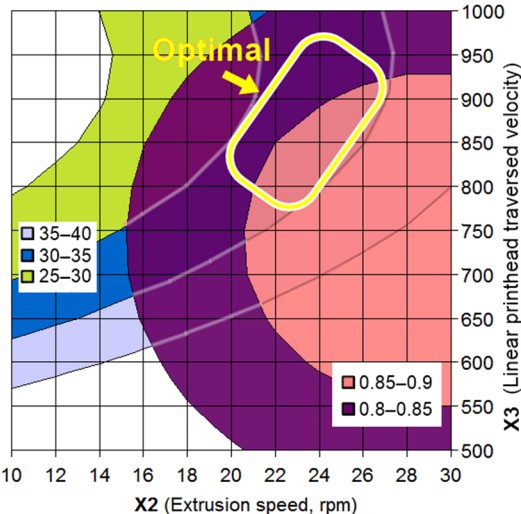

**Figure 8.** Effect of linear printhead traversed velocity and extrusion speed at the optimal layer thickness (0.8 of the nozzle diameter) on the layer thickness using the section coefficient.

The issue of the printing track kinks with an angle of 90° or more remains controversial. On the one hand, it is undesirable to reduce the productivity of the extruder to prevent the occurrence of thickening, since this will disrupt the established dynamics of the movement of the mixture and may further lead to unstable feeding for some time. On the other hand, a short-term increase in linear printhead traversed velocity in the immediate vicinity of the corner point, at a constant extruder speed, will prevent a jump in the track width at the moment of direction change, without disturbing the dynamics of the mixture in the extruder. However, a sharp change in the printing direction at high speed can cause the printer to wobble and increase the load on its units. In addition, due to the increase in the "stretching effect", rounding of the corners due to the contraction of the mixture can occur.

For the final solution of this issue, it is necessary to further clarify the allowable reduction in the amount and duration of the mixture supply, which do not affect the subsequent printing process. Another solution could be to introduce a type of track corner printing subroutine, in which the forming device at an increased speed passes a little beyond the required inflection point, and then returns to the desired track in two stages along a broken line. In this case, due to the contraction of the mixture, the actual inflection point will be in the design position.

**5. Conclusions**

1. The process of construction 3D printing by extrusion cannot be described by a linear function since the speed of rotation of the screw is not the only parameter that determines the yield of the mixture by weight per unit time. Additional operating parameters, with a constant and favorable mixture rheology, are the layer height and the linear movement of the nozzle.

2. A change in the linear movement of the nozzle can both increase the mixture yield and lead to track stretching with a decrease in parameters. According to the data obtained, the threshold value in this case is the relative layer thickness of 0.77–0.8 of the nozzle width. Up to the indicated values, with a sufficient speed of the screw, the "effect of pulling" the mixture out of the nozzle prevails due to its good engagement with the base (underlying track). The contact patch of the layers is located directly under the nozzle. A change in the direction of the vertical flow of the mixture into a horizontal one occurs both due to the inflection and, to a large extent, to shear deformations. At a higher layer height, the movement of the patch contact (layer engagement zone) lags behind the nozzle position, and a "hanging" (not fixed) section of the track appears in the inflection zone from its vertical to horizontal position. The mixture in this area is subjected to stretching with the rapid movement of the nozzle. Consequently, the mass yield of the mixture, the track width, and the use of cross-section coefficient decrease. A decrease in the linear printhead traversed velocity of the nozzle leads to a sharp increase in the track width at a low and cross-section coefficient. This is due to the fact that, at the edges of the track remote from the edge of the nozzle, the vertical adhesion force between the layers is small.

3. The response of the system to a change in the linear printhead traversed velocity and the rotational speed of screw occurs in different ways. Changing the linear printhead traversed velocity at the optimal relative height of the layer affects the width of the layer to a lesser extent, which is the most visually perceived parameter. The subsequent resumption of the previous regime occurs quickly with minimal delay and transients, since the main parameters of the mixture flow do not undergo noticeable changes.

A temporary decrease in screw speed (for example, when printing corners and sharp bends) leads to a decrease in the overall dynamics of the mixture flow and a change in its viscosity due to its inherent thixotropy. When the previous speed of rotation of the mixture is restored, the dynamics of the flow is restored with a noticeable delay. Additionally, probably, there is some limiting decrease in the flow dynamics at which the return to the initial regime will not ensure its complete restoration, without the use of additional influences (vibration or increased screw speed).

4. It is recommended to maintain the highest dynamics of the printing process provided by the capabilities of the equipment used. The farther from the limit values of the fluidity and structuring of the mixture when choosing the speed of the forming device, the less sensitive the system becomes to various disagreements during the operation of the equipment and fluctuations in the properties of the mixture. For the laboratory construction 3D printer and the molding mixture used in the work, the recommended regime, in terms of the ratio of "productivity/quality and stability of the track parameters", should be recognized as a regime with a linear printhead traversed velocity of 900 mm/min, an extruder frequency of 25 rpm, and with a relative layer thickness of 0.8 (relative to nozzle diameter). It should be understood that these parameters refer to a specific installation, which makes it necessary to develop some relative criteria for assessing the optimality of the selected regime.

5. When developing control algorithms, it is necessary to introduce special procedures for printing complex elements (corners, kinks in the trajectory, transition sections to the next layer, etc.) to compensate for undesirable relationships in the system, such as track parameters and equipment operation regimes.

**Author Contributions:** Conceptualization, M.E. and N.A.; methodology, M.E. and D.P.; software, M.E. and D.P.; validation, A.O. and V.P.; formal analysis, N.A. and N.K.; investigation, M.E. and D.P.; resources, M.E., A.O. and V.P.; data curation, D.P.; writing—original draft preparation, N.K.; writing—review and editing, N.A. and N.K.; visualization, M.E. and N.K.; supervision, M.E.; project administration, N.A. and M.E.; funding acquisition, A.O. and V.P. All authors have read and agreed to the published version of the manuscript.

**Funding:** This research received no external funding.

**Institutional Review Board Statement:** Not applicable.

**Informed Consent Statement:** Not applicable.

**Data Availability Statement:** Not applicable.

**Acknowledgments:** This study was supported by the Tomsk State University Development Programme

**Conflicts of Interest:** The authors declare no conflict of interest.

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
