# Peer review of "Influence of Equipment Operation Parameters on the Characteristics of a Track Produced with Construction 3D Printing"

_buildings, doi:10.3390/buildings12050593_

Round 1

Reviewer 1 Report

The article entitled: “Influence of equipment operation parameters on the characteristics of a track produced with construction 3D printing” is in line with the Buildings journal. The authors present interesting topic connected with using additive manufacturing technology in building industry and analysed a problems connected with this area. The organization of the article is typical for research articles. Overall, the paper is well prepared, but before publication it requires some improvements:

  • Introduction: lines 52-53 – some prototypes solution in large scale have been made, so please reformulate the sentence, because it is not true in this form. Of course, there are some problems with performance of this technology on industrial scale, for example https://doi.org/10.1016/j.jobe.2020.101833.
  • Introduction: please be more specify about “factors” (line 76) give some information about the problems with materials, for example: https://doi.org/10.1016/j.coche.2020.06.005
  • Introduction (lines 107, 116 and 129): please specify the author.
  • 2. Methods: ‘there factors’ - specify the factors.
  • Discussion: the discussion with the up-to-date literature is quite generic.
  • Author Contribution: please remove the text of the instruction given in the template.
  • Reference [13]: remove ‘1’.

Author Response

– Introduction: lines 52-53 – some prototypes solution in large scale have been made, so please reformulate the sentence, because it is not true in this form. Of course, there are some problems with performance of this technology on industrial scale, for example https://doi.org/10.1016/j.jobe.2020.101833.

Response: The sentence was reformulated. Many thanks for the recommended link to an interesting source. Link has been added to the Reference section

– Introduction: please be more specify about “factors” (line 76) give some information about the problems with materials, for example: https://doi.org/10.1016/j.coche.2020.06.005

Response: Changes have been made. Many thanks for the recommended link to an interesting source. Link has been added to the Reference section

– Introduction (lines 107, 116 and 129): please specify the author.

Response: Changes have been made.

– Methods: ‘there factors’ - specify the factors.

Response: Changes have been made.

– Author Contribution: please remove the text of the instruction given in the template.

Response: Changes have been made.

– Reference [13]: remove ‘1’.

Response: Changes have been made

Reviewer 2 Report

This manuscript presents the experimental study on the influenced of process parameters on the 3D printed construction tracks. The overall technical content was successfully elaborated with good findings supported with in-depth analysis and discussion. The paper is generally well written and has presented interesting recommendation on parameters related to a lab-scale construction 3D printer. My suggestions for improvements are as follows:

  1. Experimental method shall be enhanced with a simple flow chart to assist reading.
  2. Figures 2-5 - Please revise the caption by specifically adding (a) and (b) to differentiate the description of the two figures.
  3. Figures 3-5 – Please add a scale to the right image.
  4. Suggest changing the sub-heading for Section 2.1 to ‘Experimental setup and equipment’.
  5. Section 2.2 – please describe the used parameters of the extrusion speed and traversed velocity.
  6. Suggest moving Section 2.3 further up before Section 2.1 Methods and Equipment.
  7. Description of Table 2 is too brief. Please elaborate further.
  8. May include future work if necessary.
  9. Sufficient number of recent articles presented. Please check the first author’s name in Ref 13.

Author Response

– Experimental method shall be enhanced with a simple flow chart to assist reading.

Response: Flow chart of experimental method was added (see Figure 1b).

– Figures 2-5 - Please revise the caption by specifically adding (a) and (b) to differentiate the description of the two figures.

Response: Changes have been made.

– Figures 3-5 – Please add a scale to the right image.

Response: Changes have been made.

– Suggest changing the sub-heading for Section 2.1 to ‘Experimental setup and equipment’.

Response: Sub-heading for Section 2.1 was changed

– Section 2.2 – please describe the used parameters of the extrusion speed and traversed velocity.

Response: Changes have been made.

– Suggest moving Section 2.3 further up before Section 2.1 Methods and Equipment.

Response: Changes have been made.

– Description of Table 2 is too brief. Please elaborate further.

Response: Changes have been made.

– Sufficient number of recent articles presented. Please check the first author’s name in Ref 13

Response: Changes have been made.
